# The Influence of Rehydration on the Properties of Portland Cement-Based Materials with Low Water/Binder Ratios: A Review of Existing Research

**DOI:** 10.3390/ma16030970

**Published:** 2023-01-20

**Authors:** Liangshun Li, Yue Wang, Mingzhe An, Peiyao Yu, Xu Hou

**Affiliations:** School of Civil Engineering, Beijing Jiaotong University, Beijing 100044, China

**Keywords:** low water/binder ratio, rehydration, pore structure, influencing factors, rehydration model

## Abstract

Cement-based materials with a low water/binder ratio contain a high number of unhydrated cement particles, which implies that a rehydration reaction occurs when they encounter water again. This study aimed to explore how rehydration influences the macroscopic and microscopic properties of cement-based materials. The key study findings included that rehydration could still occur in cement-based materials after one year of hydration, and that the capacity for rehydration-induced repair or damage to cement-based materials depended on whether their internal pores could accommodate rehydration products. During rehydration, the compressive strength and porosity of the specimens were found to first increase and then decrease. The capillary water absorption coefficient decreased continuously over a rehydration period of 120 days. As the water/binder ratio rose, the rehydration rate first increased and then decreased. First, the influence of temperature on the rehydration rate was more noticeable when the water/binder ratio was below 0.3; second, whereas adding large amounts of fly ash and silica fume did not prove to be conducive to repairing and enhancing cement-based materials undergoing rehydration, adding slag and small quantities of silica fume, or alternatively compounding small amounts of silica fume and fly ash could improve the repair and enhancement effects of rehydration.

## 1. Introduction

A lower water/binder ratio and a larger amount of cement-based materials are required to manufacture high performance concrete (HPC) and ultra-high performance concrete (UHPC) compared to standard concrete, and the emergence of superplasticizers has enabled the water/binder ratio of cement-based materials to be reduced to 0.2 or even lower [1]. According to Powers’ theory, cement can only be completely hydrated when the water/cement ratio is greater than 0.42 [2], which means that a considerable amount of unhydrated cementitious materials remains inside the cement-based material after curing. Vandamme et al. [3] studied the content of unhydrated cement particles in cement-based materials with different water/binder ratios using nano-indentation technology and found that in the cement paste specimens with water/binder ratios of 0.15 and 0.20, the content of the unhydrated cement particles was 41% and 30%, respectively. Linderoth et al. [4] established that the unhydrated cement content of cement paste with water/binder ratios of 0.30 and 0.50 was 7.9% and 32.4%, respectively, after 1 year. These unhydrated cementitious materials have been shown to be capable of undergoing a rehydration reaction when subsequently encountering water, which was defined as a ‘rehydration reaction’ [5,6], and could occur not only in UHPC, but also in HPC and even in ordinary concrete. In this paper, after the 28 d hydration age of the cement-based materials, the hydration reaction that occurs when the external water enters the cement-based material and makes contact with the unhydrated cement particles is regarded as the rehydration reaction.

According to Powers’ theory [2,7], the process of cement hydration is accompanied by a decrease in total volume and an increase in solid volume, where the volume of cementitious hydration products is 2.06 times that of the unhydrated cement particles. The rehydration reaction occurs within the cement-based material after hardening, requiring additional space to accommodate the volume expansion of its hydration products, which inevitably affects the rehydration performance of cement-based materials [8,9] It plays a positive role when the space within the hardened cement-based material is sufficient. Lark et al. [10] found that the repairing effect of rehydration on the cracks of cement-based materials effectively prolonged the service life of the reinforced concrete structures. Wang et al. [11] discovered that the compressive strength of specimens after 400 freeze–thaw cycles and 1 year of re-soaking increased by 7.2% compared to samples that only underwent freeze–thaw cycles. Because hardened cement paste with a low water/binder ratio has a dense structure [12,13], its internal space is insufficient, and the stress generated by the volume expansion of the rehydration product causes damage to its internal structure. Hillermeier et al. [14] immersed HPC with a water/binder ratio of 0.3 and a compressive strength of 130 MPa at 28 days in water at 90 °C to accelerate hydration, and identified a large number of visible cracks after 7 days of immersion. This is because the expansion stress generated by the rehydration product damages the hardened cementitious material. Meanwhile, Liu [5] conducted an accelerated rehydration test on cement mortar specimens through high temperature water immersion curing at 80 °C. The samples demonstrated a significantly reduced strength after rehydration for 21 days as well as internal micro-cracks. Yang et al. [15] showed that the compressive strength of concrete with a water/binder ratio of 0.28 after rehydration at 80 °C for 7 days was 16.1% lower than that before rehydration. Elsewhere, research by Pushpalal et al. [16] found that the compressive strength of high alumina cement/phenolic resin composites with minimal hydration decreased by 9% after being placed in an outdoor environment for one year, which they ascribed to the rehydration of unhydrated cement.

Many cement-based materials with a low water/binder ratio are subject to significant rehydration reactions due to the presence of large amounts of unhydrated cementitious materials within them. Notwithstanding the above, few studies have been conducted to date, focusing on the rehydration of cement-based materials with a low water/binder ratio and on the effect of rehydration on their microscopic properties, as this effect is mainly reflected in their macroscopic properties. There is no consensus on the long-term performance of the evolution law in controlling the adverse effect of the rehydration reaction on cement-based materials with a low water/binder ratio, and there lacks an effective damage risk evaluation and control method. Since the rehydration process of cement-based materials with a low water/binder ratio and its mechanism remain unclear, it is necessary to study the mechanism of performance change in cement-based materials with a low water/binder ratio under the rehydration effect. To this end, this study was conducted. First, the effect of rehydration on the long-term performance of cement-based materials with a low water/binder ratio was systematically analyzed, and the repair of cracks, mechanical properties, and permeability changes in cement-based materials under rehydration were systematically analyzed. Second, the influencing factors of rehydration were determined, aiming to provide theoretical guidance for the long-term performance stability of cementitious materials and to clarify how to effectively control the adverse effect of rehydration on cement-based materials with a low water/binder ratio. Finally, a systematic solution was provided to suppress the deterioration caused by rehydration on cement-based materials.

## 2. Influence of Rehydration on Macroscopic Properties of Cement-Based Materials

Rehydration only occurs when the following conditions are met [8]: first, the cement-based materials must contain unhydrated cement particles, and water must be able to penetrate these during the service life of the structure. Research to date has focused on the changes in the durability index [17], the behavior of chemically combined water [18], and on the mechanical properties and capillary water absorption [19] in the long-term service of cement-based materials with a low water/binder ratio. The details are described as follows.

### 2.1. Effects of Rehydration on Cracks

The phenomenon of rehydration was first identified in cement-based structures serving in water such as water storage constructions [20]. Rehydration was shown not only to have the capacity to repair leakages of concrete structures [21] and reduce the penetration of aggressive ions into cement-based materials [22,23], but also to significantly prolong the service life of concrete structures [10]. Yan et al. [24] found that cement-based materials could remain in working order with cracks during their service. They established that those with low water/binder ratios displayed a certain ability to self-repair while operating with cracks. Research by Reinhardt and Jooss [25] confirmed that (i) rehydration could repair cracks in concrete; (ii) the smaller the crack, the easier the repair work; and (iii) the ability to self-repair increased with the temperature. The repair mechanism of cement-based materials under rehydration is shown in Figure 1. Other studies [26] ascertained that the narrow width of cracks occurring within engineered cementitious composite (ECC) materials conferred these with an evident capacity for self-restoration by rehydration, an equally manifest feature of fiber-reinforced cement-based materials with a dense interior and small crack widths [27]. These latter findings were corroborated by Reinhardt and Jooss’s research [25]. Elsewhere, some scholars combined experimental and simulation methods to study the process of new crack propagation during the repair process of cement-based materials [10,28]. For instance, Aldea et al. [29] found that the maximum crack width capable of self-repair by rehydration within cement-based materials was 200–300 μm, and that the greater the initial crack width, the slower the restoration speed. A study by Tomczak and Jakubowski [30] showed that whereas the maximum crack width in cement-based materials capable of self-repair within 2 months of rehydration was 460 μm, the internal unhydrated cement particles of the cement-based materials that had undergone a rehydration period of 20 months still retained their crack repairing ability. The maximum repairable crack width under this latter condition was established as 388 μm, with the repair rate decreasing with age. Meanwhile, Saharan et al. [31] established that while the strength of concrete specimens with a water/binder ratio of 0.35 decreased by 27% under preloading, after soaking them in water for curing, the previously identified cracks were now repaired due to the rehydration of unhydrated cement particles within the concrete. The strength of the samples had recovered by 20% after rehydration for 30 days. According to Ahn and Kishi [32], the unhydrated cementitious materials within cement-based materials restored cracks of only 150 μm in width by rehydration after 3 days, against 7 days for cracks of 160–220 μm in width. Elsewhere, it was found by scanning electron microscope (SEM) and computed tomography (CT) techniques that cracks with a width below 50 μm could be completely repaired by rehydration after re-curing for 33 days [33]. Guo et al. [34] reported that the repair mechanism of the rehydration of cement-based materials mainly consisted of cracks of less than 20 μm in width becoming filled. This finding was upheld by Yang et al. [35], whose research showed that cement-based materials repaired cracks by rehydration under the action of dry/wet cycles, the maximum repairable crack width was 150 μm, and that optimal restoration occurred when the crack width was less than 50 μm. In a different approach, Benny et al. [36] studied the repairing effect of ultrasonic methodology on the rehydration of cement-based materials and established that narrow hairline cracks (<10 μm wide) could be restored within 6 days of exposure to external environments, and that partial or complete repair could be achieved for 20–30 μm wide cracks after 6 days of intermittent rainfall, while cracks of 40–75 μm wide could be partially restored after 3 weeks of exposure to outdoor conditions. Meanwhile, the results of Liu et al. [37] revealed that cracks with a width of 150 μm were completely closed after 120 days of rehydration, and that the addition of sulfate solution (Na_2_SO_4_) could accelerate the crack closing process. The maximum crack width that can be repaired by cement-based materials under rehydration is summarized in Table 1. In summary, the above findings collectively indicated that (i) the repairing effect of rehydration on cement-based materials was the best when cracks were less than 50 μm wide; (ii) the widest restorable crack identified to date was 460 μm; and (iii) the repairing effect was influenced by factors such as temperature, water/binder ratio, and the addition of fiber.

A significant finding highlighted by An et al. [38,39] was that although the rehydration process could repair cracks, it also presented the potential to induce cracks when insufficient space was available for the expansion of hydration products within cement-based materials. This was confirmed when An et al. [11] immersed UHPC specimens with a water/binder ratio of 0.2 in water for rehydration for 2 years. They found that the rehydration of the unhydrated cementitious materials within them resulted in micro-cracks developing between the UHPC and the steel fiber contact surfaces, in turn reducing the compressive strength of the samples by 14.5%. In another experiment, Hillermeier and Schroeder [14] accelerated the hydration of HPC with a water/binder ratio of 0.3 by a water immersion method at 90 °C. After curing the specimens for 7 days, visible cracks were observed on their surfaces. After curing other cement paste specimens with a water/binder ratio of 0.2 by high temperature steam, no obvious cracks were noted on the sample surfaces. However, after subsequently soaking them in water at 20 °C for 14 days, obvious cracks could be detected on the specimen surfaces in different crack forms such as mesh, through-type, figure-of-eight, and X-cross type after soaking the specimens with different water/binder ratios at room temperature for 90 days [6]. An et al. [9] established that in the early stage of rehydration, the pores within the UHPC matrices with low water/binder ratios were filled with rehydration products that may have had a reinforcing effect. In the later stage of rehydration, however, the internal pores proved inadequate to accommodate new hydration products, thus generating internal expansion stress that eventually led to the formation of new cracks. As shown in Figure 2, when the cement-based material was rehydrated for 56 d, the rehydration product filled the pores, and after 180 d of rehydration, the pores were not enough to accommodate the rehydration product, which eventually led to the generation of new cracks. The above findings can be summarized by stating that whether or not cracks are generated by rehydration within structures depends on whether or not there is space for rehydration products within these.

### 2.2. Effects of Rehydration on Mechanical Properties

It can be intuitively accepted that changes in the compressive strength reflect the effect of rehydration on the concrete properties. In the present study, compressive strength was one of the main indicators used to characterize the influence of rehydration. In studying how rehydration could repair cement paste specimens, Lauer and Slate [41] established that the process increased their tensile strength perpendicular to the crack direction, while Dhir et al. [42] likewise found that rehydration enhanced the strength of cement-based materials. Igarashi et al. [43] studied the properties of cement paste specimens after rehydration in water at 20 °C and found that with 10% silica fume admixture and a water/binder ratio of 0.24, their mechanical properties decreased by 9.8% after 179 days of rehydration compared to 89 days of rehydration. Elsewhere, Feng et al. [44] and Ge et al. [45] investigated the effect of rehydration on the strength of concrete with a low water/binder ratio by accelerating hydration by water immersion in high-temperature water. The test results showed that as the duration of accelerated rehydration extended, the concrete strength trend was first to increase and then to decrease, with a significant reduction in the latter period. In addition, Roz et al. [46] studied the mechanical properties of wood fiber cement, and the results showed that the presence of wood fibers in the cement-based materials doped with wood fibers would not be conducive to improving the mechanical properties of cement-based materials under rehydration. This is mainly due to the reversible damage caused by water to the hydrogen bond between the cellulose fibers and the cement-based matrix, which eventually leads to a significant decrease in the flexural properties of cementitious materials. Yang and Guan [15] reported the presence of a large amount of unhydrated cement in concrete with a low water/binder ratio concrete, the rehydration of which led to concrete damage in the form of a significant decline in the compressive strength and durability.

In order to more systematically analyze the effect of rehydration on the compressive strength of cement-based materials, the compressive strength of cement-based materials with low water/binder ratios at corresponding ages of rehydration in the existing research results in the literature [5,6,19,40,43,44,45,47,48,49,50,51,52,53,54,55,56,57,58,59,60,61] was processed. The curing method of the specimen was to perform standard curing first, and then to accelerate the hydration by water immersion curing. Here, the compressive strength at the age of 28 days was normalized as 1, and the results are shown in Figure 3.

Figure 3 shows that up to 270 days of hydration, the compressive strength increased with duration, albeit the growth rate decreased continuously during this phase. After 270 days of hydration, the compressive strength could be observed to have consistently decreased, whose stage was interpreted as the internal damage phase to the cementitious materials due to the solid expansion-induced stress during the hydration of cement-based materials. The normalized compressive strength of cement-based materials was reduced by approximately 15.21% under rehydration. Current studies on the rehydration of cement-based materials for more than 150 days are scarce. In order to fully study the effect of long-term rehydration on the properties of cement-based materials, in the future, it is necessary to investigate the performance changes of cement-based materials after the rehydration age exceeds 150 days. The normalized compressive strength was fitted segmentally according to a logarithmic function for the hydration period of 0–270 days (*R*^2^ = 0.77), and thereafter linearly after the 270th hydration day (*R*^2^ = 0.72). The corresponding curves after fitting are shown in Formula (1):(1)$$\sigma_{\mathrm{nc}}=\left\{\begin{array}{l} 0.619-0.123\ln(t-0.922),t\leq 270\\ 1.815-0.0017t,t>270 \end{array}\right.$$
where *σ*_nc_ is the normalized compressive strength and *t* is the hydration age.

The increased rehydration degree of cement-based materials was represented by the increased volume of combined water and the effects of rehydration on these could conversely be characterized as the impact of rising hydration on the compressive strength. The relationship between incremental combined water and compressive strength growth could then be analyzed. Given that most of the currently available studies have only characterized the amount of chemically combined water within a 90-day hydration period of cement-based materials, the relationship between the incremental rise in the combined water and compressive strength of cement-based materials within 90 days of rehydration reported in the literature [5,45,55,56,57,60,61] was analyzed in this study, as illustrated in Figure 4. Within 90 days of hydration, as rehydration progressed, the compressive strength of the cement-based materials with a low water/binder ratio continued to increase, suggesting that the rehydration effect mainly consisted in the internal pores becoming filled by the rehydration products. The internal density of the samples thus rose, as did the compressive strength, revealing in detail the effect of rehydration on porosity. This finding is consistent with the law of the compressive strength changes alluded to in Figure 3 above, where the hydration period was below 90 days.

A polynomial function was used to fit the relationship between incremental combined water and compressive strength increases, from which the relationship between the incremental compressive strength and chemically combined water content of cementitious materials within 90 days of rehydration was obtained, as shown in Formula (2):(2)$$\sigma_{c}=0.586W_{\mathrm{nt}}{}^{2}+5.556W_{\mathrm{nt}}+4.791$$
where *σ*_c_ is the compressive strength and *W*_nt_ is the chemically combined water content.

### 2.3. Effect of Rehydration on Permeability

Whereas water migration in porous materials primarily involves capillary water absorption, diffusion, and infiltration under pressure gradients, that in cement-based materials mainly consists of capillary water absorption [62]. The capillary water absorption characteristics of cement-based materials can readily characterize both their impermeability and pore structure, which are highly important indicators for evaluating the durability of cement-based materials. Capillary water absorption coefficients are therefore generally used to indicate the capillary water absorption properties of cement-based materials. An et al. proposed a preload damage treatment method for cement-based materials [63]. Zhu et al. [64] analyzed the variations in the performance of ECC materials damaged by preloading after freeze–thaw cycles. Not only were the cracks generated by preloading the ECC materials found to have been repaired by rehydration under the action of freeze–thaw cycles, but the capillary water absorption coefficient of the ECC specimens with a preloaded tensile strain below 1.5% was close to that of the undamaged ECC specimens after rehydration during the freeze–thaw cycles. Since current research on capillary water absorption coefficients is mainly focused on a 120-day hydration period, in order to ensure the validity of the data, only the capillary water absorption coefficients within the 120-day hydration period reported in the literature [19,61,65,66,67,68,69,70,71,72] were analyzed in this study. In addition, the capillary water absorption coefficient of cement-based materials with low water/binder ratios was selected for the experiment, and the capillary water absorption coefficients at 28 days was normalized as 1, as shown in Figure 5.

Figure 5 shows that the capillary absorption coefficient decreased continuously as the hydration period extended, implying that the impermeability, durability, and density of the cement-based materials continued to improve in line with the continuous progress of rehydration during the 120-day hydration period. The normalized capillary absorption coefficient of cementitious materials with the hydration age of 120 d was reduced to 18.65% of the initial state. The relationship between the normalized capillary water absorption coefficient and hydration age was fitted by a power function (*R*^2^ = 0.75), the result of which is shown in Formula (3):(3)$$S_{\mathrm{nm}}=4.24-2.31t^{0.109}$$
where *S*_nm_ is the normalized capillary water absorption coefficient and *t* is the hydration age.

The changes in the macroscopic properties of cement-based materials under rehydration are summarized in Table 2. The current capillary water absorption test results are often largely dispersed. In future study, it is recommended that low-field nuclear magnetic resonance (NMR) and X-ray computed tomography (X-CT) methods are used to detect the capillary water absorption and capillary water absorption depth, so that the accuracy of characterizing the permeability of cement-based materials can be improved.

## 3. Influence of Rehydration on Microscopic Properties of Cement-Based Materials

The effects of rehydration on the properties of cement-based materials is mainly reflected in the repair or damage caused to the interior of cement-based materials by rehydration products. This process is accompanied by and ultimately manifested as associated changes in microscopic properties such as the internal pore structure of cement-based materials. The influence of rehydration is mainly reflected in the type and content of rehydration products. At present, the methods commonly used for the qualitative analysis of rehydration products are thermogravimetric analysis (TGA) [73] and X-ray diffraction (XRD) [74], while infrared spectroscopy is often used for the quantitative analysis of the same. The following subsection provides a detailed analysis of the effects of rehydration on the microscopic properties of cement-based materials.

### 3.1. Rehydration Products

Various types of rehydration products have been the subject of much research by many scholars, who have established that hydration products generated by the rehydration of unhydrated cementitious materials could significantly influence the microscopic properties of, and ultimately the long-term performance changes in, cement-based materials. Huang [22] studied the hydration products of cement paste with a water/binder ratio of 0.3 in a water immersion environment where cracks were found to have been repaired and the main components of the restorative product were crystalline calcium hydroxide (Ca(OH)_2_), gelatinous C-S-H gel, and calcium carbonate, and 80% of the rehydration product was reported to be crystalline Ca(OH)_2_. Rong’s study [75] found that the unhydrated cement particles of cement-based materials with low water/binder ratios were wrapped in ultra-high-density C-S-H gel, and the hydration rate was slower. The cement-based materials with mineral admixtures contained very low levels of Ca(OH)_2_, which continued to reduce the longer the rehydration process was pursued. Meanwhile, Taylor et al. [76] found that 20-year aged concrete could still rehydrate, and the Ca(OH)_2_ content decreased as rehydration proceeded. The Ca/Si ratio of C-S-H at 20 years was lower than that of C-S-H at 14 months of hydration. Jacobsen and Sellevold [77] rehydrated HPC specimens after freeze–thaw damage by immersing them in water for 90 days and found that the newly-generated products were C-S-H gels, Ca(OH)_2_, and AFt.

In other studies, Edvardsen [21] established that CO_2_ could be dissolved in water to form CO_3_^2−^, which invaded the inside of the specimens through cracks. Once Ca^2+^ and CO_3_^2−^ were supersaturated, CaCO_3_ was generated in the cracks, while Sisomphon et al. [78] identified a significant amount of CaCO_3_ after testing the rehydration products at the surface part of the cracks. Parks et al. [79] and Choi et al. [80] studied the autogenous repair of cracks in concrete structures and produced “simulated cracks” by cutting concrete into two halves and then fixing the “crack” width using polyethylene spacers. They suspended these specimens in 2 L containers with water, further reporting that when the pH value of the water environment was 9.5, significant repair could occur, and that placing the specimens in a saturated Ca(OH)_2_ solution with added CO_2_ microbubbles could enhance the self-repair capacity of the specimens. Guo et al. [34] proposed that the self-repair process of cement-based materials was primarily achieved by the continuous hydration of unhydrated cement particles, the pozzolanic reaction of fly ash particles, and the precipitation of CaCO_3_ crystallization. Meanwhile, Zhu et al. [64] found that rehydration caused white matter to form around the ECC cracks, the extent of self-repair differed under different preload stresses, and that increasing the preload stress enhanced the ability of the samples to self-repair. XRD analysis of the white material confirmed that the chemical composition of the rehydration products was CaCO_3_ and Ca(OH)_2_. Kan et al. [81] examined rehydration products by drilling into the crack and other methods and discovered that these were fibrous C-S-H gels and crystalline CaCO_3_, and that their types were correlated with the crack width, whereas the main rehydration product identified in the presence of the 15 μm crack widths was C-S-H gel, and those with crack widths of 30 μm were C-S-H gel and CaCO_3_, with fewer rehydration products with crack widths of 50 μm. Liu et al. [37] preloaded ECC specimens at 1% strain and exposed them to sulfate and sulfate-chloride solutions, followed by the analysis of rehydration products by EDX spectroscopy, which indicated that the rehydration products were mainly C-S-H gel, Ca(OH)_2_, ettringite, and CaCO_3_, and that sulfate could promote the formation of ettringite. Elsewhere, Jacobsen et al. [82] studied concrete specimens before and after freeze–thaw cycles and concluded that the specimens continued to undergo a damage–restoration process during the freeze–thaw cycles. They contended that during the rehydration, the unhydrated cement particles within the concrete continuously underwent a rehydration reaction with the water following internal thawing. The C-S-H gel generated gradually repaired the internal damage caused by freezing and thawing, thus resulting in a significantly increased resonance frequency of the concrete during thawing. Schlangen et al. [83] re-cured specimens with cracks in water and observed that new C-S-H gels had formed in the cracks under rehydration, interpreting this to indicate that the rehydration of unhydrated cementitious materials had produced C-S-H gels. The summary of the rehydration products of the current studies is shown in Table 3, The specific morphology of these major rehydration products observed under scanning electron microscopy is shown in Figure 6.

In summary, the main rehydration products according to the literature were fibrous C-S-H gel and crystalline CaCO_3_, with the potential concurrent generation of a certain amount of Ca(OH)_2_ and AFt. The current commonly used method for detecting rehydration products is XRD. The XRD results obtained can be combined with transmission electron microscopy to analyze the type of rehydration product and with thermogravimetry to analyze their content, thus achieving a more accurate quantitative analysis.

### 3.2. Effect of Rehydration on Pore Structure

Pore structure is an important component of the microstructure of cement-based materials [84], and the total porosity and pore structure distribution of cementitious materials are often analyzed by mercury intrusion porosimetry (MIP) [85]. As previously indicated, rehydration affects the internal pore structure of cement-based materials, as established by Yu et al. [86]. They studied the changes in the pore structure of cement mortar during a rehydration process. They also reported that, first, the total porosity of the cement mortar samples after standard curing for 209 days decreased continuously with the increasing rehydration time; and second, the decline rate in porosity was reduced from 98 days of hydration. In order to systematically study the effect of rehydration on porosity, the porosity of cement-based materials with low water/binder ratios, low mineral admixtures, and under similar long-term maintenance conditions at different hydration stages cited in the literature [5,50,56,58,59,61,72,75,87,88,89,90,91,92,93] were analyzed in the present study. To minimize the influence of other factors, the porosity was normalized by considering the porosity at 28 days of hydration as 1. The changes in the porosity of cement-based materials at the above hydration age are shown in Figure 7. The porosity decreased rapidly in the early stage of hydration, after which the porosity reduction rate slowed down. After the hydration age had reached 270 days, the porosity was found to have increased again.

The research available to date on the rehydration of cement-based materials has mainly focused on rehydration ages of up to 120 days, so there are relatively sufficient sample data on rehydration less than 120 days. In contrast, studies on rehydration for over 120 days are relatively scarce, resulting in few data relating to pore structure and limited fitting results for rehydration ages of 270 days. In light of the above, the rational function was used in the present study to fit the relationship between the porosity and hydration age below 270 days (*R*^2^ = 0.84). The relationship between the porosity after 270 days of hydration and the hydration age was fitted with a linear function (*R*^2^ = 0.99), and the results are shown in Formula (4):(4)$$\left\{\begin{array}{l} P_{n}=\frac{3.146+0.824^{t}}{0.574+t},t\leq 270\\ P_{n}=0.534+0.00168t,t>270 \end{array}\right.$$
where *P*_n_ is the normalized porosity and *t* is the hydration age.

From Figure 3 and Figure 7, it can be seen that while the porosity of cement-based materials decreased continuously with rehydration over the age of 0–270 days, the internal compactness increased continuously, which in turn indicated that the compressive strength increased continuously. The porosity after 270 days of hydration once again showed a rising trend consistent with the change law of compressive strength. Although the currently available study results on hydration duration-related porosity changes are limited, the present authors believe that a sound analysis of the results available could establish a link between the macroscopic and microscopic properties of cement-based materials as a whole. We therefore regard it as essential in the future to study the properties of cement-based materials above 120 days of rehydration by conducting relevant tests. Currently, the pores of cement-based materials are often tested and analyzed by MIP, which requires the destruction of the specimen. In order to ensure the continuity of pore structure changes during the rehydration of cement-based materials, the low-field NMR method can be used to test the pore structure of the cement-based materials. The results obtained by low-field NMR can be compared with those obtained by the MIP method to characterize the pore structure of cement-based materials under rehydration more clearly.

## 4. Influence of Different Factors on Rehydration

The effect of rehydration on cement-based materials can be characterized by the variable of chemically combined water, which can have an effect on rehydration in complex ways such as the composition of cementitious materials and the state of maintenance of the structure. Identifying the effects of these factors on the rehydration of cement-based materials is critical for exploring the mechanical properties and even the durability of cement-based materials. In this study, the influence of diverse factors on the rehydration of cementitious materials such as the water/binder ratio, fly ash and silica fume admixture, and rehydration temperature conditions were investigated through literature research and analysis, with the view to establishing a law of influence of different factors on rehydration.

### 4.1. Water/Binder Ratio

Because changes in water/binder ratios have been shown to significantly affect the rehydration of cement-based materials with low water/binder ratios, the data on variations in combined water content under different water/binder ratios during rehydration cited in the relevant literature [6,40,45,60,61] were analyzed in the present study. In order to characterize the influence of water/binder ratios on the rehydration rates, the average daily alterations in chemically combined water were obtained by dividing the chemically combined water variations by the corresponding number of hydration days for comparison. The average daily increments in the chemically combined water corresponding to different water/binder ratios in the literature were then plotted in a histogram, as shown in Figure 8.

It can be seen from Figure 8a,b that when the water/binder ratio was between 0.15 and 0.2, as it rose, the average daily growth rate of the chemically combined water increased slightly. Figure 8c shows that the water/binder ratio significantly affected the rehydration of cement-based materials, and that rehydration not only exists in cement-based materials with low water/binder ratios, but also in cement-based materials with a water/binder ratio of 0.4. There are still a large number of unhydrated cement particles inside the cement-based materials after the completion of curing, and the rehydration reaction can still occur after the entry of water in the latter stage. The statistical results in Figure 8 shows that the average daily variation rate of the chemically combined water generally rose first, and then reduced. The main factors controlling the rehydration rate were believed to be the water penetration and the unhydrated cement content. As the water/binder ratio continued to decrease, the interior of the cement-based materials became denser and less conducive to water penetration. However, the lower the water/binder ratio, the more unhydrated cement particles were available for the rehydration reaction, the rate of which was determined by the combined effect of the two. The turning points of the influence of the water/binder ratio on rehydration (generally between 0.2 and 0.3) varied according to the variations in cement types and rehydration temperatures. In addition, as can be seen from the literature [45], the rehydration rate of cement-based materials with the same water/binder ratio varied greatly at different rehydration temperatures, indicating that the water temperature in the latter stage of immersion hydration significantly influenced the rehydration process. It therefore proved necessary to study the impact of rehydration temperatures on rehydration.

### 4.2. Rehydration Temperatures

As indicated in the above literature, temperature variations during the accelerated hydration process were found to significantly affect the rehydration of cement-based materials mainly in two respects: first, the cement hydration rate was shown to increase significantly with the increasing temperature; and second, the compactness of the hydration product increased with the rise in temperature, which impeded water penetration into the cement-based material in the latter stage of rehydration.

In the present study, the literature [40,45,50,55] data on the chemically combined water content within 90 days of rehydration under constant water/binder ratios but variable rehydration temperatures were analyzed. Average daily chemically combined water increments were used to determine the rehydration rates of cement-based materials at different temperatures, as shown in Figure 9.

Figure 9a shows that after 90 days of rehydration, the average daily increments of chemically combined water at rehydration temperatures of 40 °C, 60 °C, and 90 °C increased by 27.20%, 34.49%, and 53.93%, respectively, compared with that at a rehydration temperature of 20 °C for cement-based materials with a water/binder ratio of 0.3. Figure 9b indicates that after 90 days of rehydration for cement-based materials with a water/binder ratio of 0.2, the average daily increments of chemically combined water at rehydration temperatures of 30 °C, 40 °C, 60 °C, and 90 °C increased by 31.51%, 60.06%, 74.20%, and 76.39%, respectively, compared with that at a rehydration temperature of 20 °C. Figure 9c illustrates that the average daily increment of chemically combined water at a rehydration temperature of 80 °C was 20.88 times higher than that at a rehydration temperature of 20 °C for cement-based materials with a water/binder ratio of 0.2 after 28 days of rehydration. As shown in Figure 9d, after 90 days of the rehydration of cement-based materials with water/binder ratios of 0.15, 0.17, 0.18, 0.20, and 0.30, respectively, the average daily increment of chemically combined water at a rehydration temperature of 60 °C was 3.69%, 66.37%, 16.09%, 13.35%, and 0.70% higher than that at a rehydration temperature of 20 °C. The cement-based material with a water/binder ratio of 0.2 had the largest increase in the average daily increment of chemically combined water under rehydration. This indicates that for cement-based materials with a water/binder ratio of 0.2, more attention should be paid to the adverse effect of the rehydration reactions on them. It can be seen from Figure 9 that the rehydration rate of cement-based materials increased with the rise in temperature, irrespective of the water/binder ratios. Figure 9d illustrates that when the water/binder ratio was very low (0.15), the cement-based material had a very high density, making it difficult for water to penetrate the cement-based material. The acceleration effect of temperature on rehydration was thus not obvious due to the lack of water. In contrast, as the water/binder ratio increased, the accelerating effect of the raised temperatures on rehydration became apparent. Although the amount of unhydrated cement particles that could be provided in the cases of cement-based materials with higher water/binder ratios cured for a period of time was relatively small, it could result in a low rehydration rate. These findings suggest that temperature is an important factor affecting the rehydration of cement-based materials with low water/binder ratios.

### 4.3. Mineral Admixtures

Tittelboom et al. [94] studied the effects of cementitious material types on the rehydration of cement-based materials and found that adding mineral powder or fly ash could accelerate their rehydration, and that the accelerating effect of mineral powder was more significant. The maximum crack after rehydration of the cement-based material mixed with slag was half of the crack width after the rehydration of the cement-based material mixed with fly ash. Hee et al. [95] studied the properties of Ferro-nickel slag (FNA)-doped cement-based materials by using XRD, TG analysis, and MIP. They found that the average pore size of the FNA-doped cement-based materials was reduced, and that the rehydration products produced by the reaction between calcium hydroxide and internal moisture from the decomposition of C-S-H gels at high temperatures were different from those produced by the rehydration products of cement-based materials not doped with FNA and could have better refractory properties. Yang et al. [96] investigated the inhibitory effects of mineral admixtures on rehydration, whose results showed that adding a mixture of fly ash and slag could effectively inhibit the damage caused by rehydration.

Limited studies currently exist as to the extent to which mineral admixtures enhance the rehydration process of cement-based materials. In the present study, the data on the compressive strength available in the literature [48,50,53,55,60,61] were analyzed, with the compressive strength at 28 days of standard curing being selected as the compressive strength of rehydration at 0 days. After obtaining the data on the increased compressive strength of cement-based materials with different types and amounts of mineral admixtures after rehydration for a period of time, these were divided by the corresponding compressive strength of rehydration at 0 days to determine the compressive strength growth rate, whose results are shown in Figure 10.

Figure 10a shows that when the ratio of metakaolin was 15% or less, the compressive strength of the cement-based materials with standard curing for 28 days could be improved. This may be because the metakaolin powder accelerated the early hydration of the cement [97], thereby improving the initial strength. However, the increase rate in the strength of cement-based materials was found to decrease under rehydration, indicating that the addition of metakaolin reduced the repair and enhancing effect of rehydration on the cement-based materials. In contrast, whereas when the limestone powder dosage was 30%, the compressive strength of the cement-based materials after standard curing for 28 days decreased, and the compressive strength after rehydration for 28 days was shown to have improved compared to the cement-based material samples without mineral admixtures, indicating that limestone powder could enhance the repair and enhancing effect of rehydration on cement-based materials. The results furthermore established that the compounding of metakaolin and limestone powder could improve the compressive strength of the specimens of cement-based materials at 28 days of rehydration and beyond.

As presented in Figure 10b,c, the initial compressive strength of the cement-based materials with a water/cement ratio of 0.17 and mixed with either 5% silica fume alone or with 5% silica fume compounded with 15% fly ash, was greater than that of the samples with no admixture. Interestingly, however, the initial compressive strength of the cement-based materials mixed with either 10% silica fume alone or 10% silica fume compounded with 20% fly ash proved to be lower than that of the samples with no admixture. In addition, the compressive strength of the cement-based materials with mineral admixtures after 90 days of rehydration remained lower than that of the specimens without the mineral admixture. Therefore, the repair and enhancing effect of rehydration on cement-based materials with a water/cement ratio of 0.17 was reduced when the ratio of silica fume and fly ash was large. For cement-based materials with a water/binder ratio of 0.2 and a low proportion of silica fume, the compressive strength exceeded that without any mineral admixture in every rehydration stage. It was therefore concluded that the addition of silica fume improved the repair and enhancing effect of the rehydration of cement-based materials. In summary, the influence of silica fume and fly ash on the rehydration of cement-based materials also proved to be closely related to their water/binder ratio.

Figure 10d,e illustrates how the initial and final compressive strengths of cement-based materials were reduced by adding fly ash, indicating that the addition of fly ash was not conducive to the repair and enhancing effect of cement-based materials under rehydration. In the case of cement-based materials with high water/binder ratios and with added fly ash, the rate of increased compressive strength under rehydration increased with the increasing fly ash content, and the rate of the increased compressive strength under rehydration of the samples with low water/binder ratios reduced as the proportion of fly ash rose. Figure 10d shows that for cement-based materials with a water/binder ratio of 0.4, when fly ash was added, there was still unhydrated cementitious material inside the specimens at the completion of curing. Rehydration would occur subsequently, which will influence the performance of cementitious materials. Finally, as can be seen from Figure 10f, that while the initial and final compressive strengths of the cement-based materials were reduced after adding fly ash, the later strength of the specimens with high water/binder ratios increased significantly. This finding is consistent with those of other scholars. While the initial compressive strength of cement-based materials was found to have decreased after adding slag, it proved to be higher after 32 days of rehydration than that of the samples without slag. It was thus concluded that although the addition of fly ash did not prove to be conducive to the repair and enhancing effect of cement-based materials under rehydration, the addition of slag did. The effects of different factors on the rehydration of cement-based materials are summarized in Table 4.

## 5. Research Status of Rehydration Models

Existing studies on the extent of cement hydration have mostly employed either the hydration heat method [89] or the chemically combined water method [58]. Whereas the rehydration process usually occurs in the latter stages of the hydration of cement-based materials, the hydration heat method has proven to be effective in characterizing the early hydration data. In this early stage, the heat released by cement hydration is low, resulting in a slow heat release curve with no obvious variation trend. Reliance on the hydration heat method therefore implies incurring a large margin of error, in addition to which the heat of hydration produced during the rehydration process cannot be tested when it occurs in a water environment. The method of chemically combined water is therefore often instead utilized to investigate the extent of cement hydration during the rehydration process.

### 5.1. Effects of Rehydration on Chemically Combined Water

The results of the chemically combined water tests in the literature [5,6,19,45,55,56,57,60,61,92,98,99,100] conducted on cement-based materials with low water/cement ratios were selected here for analysis (W/B ≤ 0.38). According to various studies, the samples of cement-based materials with low water/binder ratios were reportedly first subjected to standard curing for 28 days, followed by an accelerated hydration process using water immersion. The various data were normalized, the chemically combined water content corresponding to the 28 day of standard curing hydration of the specimens was 1, and the relationship between the normalized chemically combined water and hydration age was analyzed, as shown in Figure 11.

Figure 11 shows that in the early stage of hydration, the chemically bound water content of the cement-based materials rose rapidly, after which the increase rate slowed down. Despite this observation, the volume of chemically bound water was shown to still have continued to increase after the hydration age had reached 388 days, indicating that in these experiments, the rehydration of cement-based materials with low water/binder ratios was a continuous process. The relationship between the normalized chemically bound water content and hydration age was fitted using a logarithmic function (*R*^2^ = 0.83), and the relationship between the chemically bound water content of the cement-based materials and their hydration age was obtained as shown in Formula (5):(5)$$W_{\mathrm{nt}}=0.661-0.106\ln\left(t-0.476\right)$$
where *W*_nt_ is the chemically bound water content of the cement-based material at the hydration age *t*; *t* is the hydration age; and *t* ≥ 0.5 in the present case.

The results in Figure 11 show that cement hydration is a long-term process in the early stage of which the hydration rate accelerates. However, this stage is often absent in the case of rehydration processes, hence the existing hydration model is no longer applicable, which implies that a new model is required to plot the rehydration stages of cement-based materials. Current methods for testing the hydration of cement-based materials often use the method of testing the chemically combined water content, and the test samples may have large errors due to different sampling locations. The relationship between resistivity and hydration can be used in subsequent studies to achieve a nondestructive method to test the hydration of cement-based materials.

### 5.2. Rehydration Models Applicable to Different Conditions

Further cement hydration models have been proposed over recent times including by Krstulović and Dabić [101], who proposed that the hydration of cementitious materials could be divided into three stages: (i) crystalline nucleation and crystal growth; (ii) phase boundary reaction; and (iii) diffusion. Yan and Zheng [102] then modified the Krstulović–Dabić model [101] and obtained a *dα*/*dt* curve that could better simulate the actual hydration rate of cementitious materials. Elsewhere, many cement hydration models developed by Maekawa et al. [103], which took into account parameters such as cement chemical components, cement fineness, water/binder ratio, and curing temperature, have proven to be effective in characterizing cement hydration. However, it is critically important to bear in mind that rehydration often occurs in the latter stages of the hydration of cement-based materials, and that the hydration characteristics at this time are quite different from those of the initial hydration stage, making it difficult to propose a rehydration model.

Based on the diffusion process in the differential equation of the three processes of the hydration kinetic process they described, Wang et al. [6] modified the original water permeability coefficient and established a rehydration model for situations when the rehydration of cement-based materials occurred in a water environment. Their results indicate that the rehydration rate was much lower than the hydration rate at the start of cement hydration, as shown in Formula (6):(6)$$\left\{\begin{array}{l} d\alpha /dt=3/2k_{r}{\left(1-\alpha \right)}^{2/3}/[1-{\left(1-\alpha \right)}^{1/3}D_{e0}\ln(1/\alpha )/r_{0}]\\ k_{r}=k_{r20}\exp\left(-\beta \left(\frac{1}{T}-\frac{1}{20}\right)\right) \end{array}\right.$$
where *α* is the degree of hydration; *t* is the rehydration time; *k_r_* is the rehydration rate considering the influence of temperature, silica fume and curing conditions, is an undetermined constant; *k_r_*_20_ is the value of *k_r_* at 20 °C; *r*_0_ is the radius of unhydrated cement particles; and *D*_e0_ is the effective diffusion coefficient of water within the C-S-H gel before rehydration commences.

The present authors deemed this rehydration model suitable for predicting the degree of hydration of cement–microsilica fume systems, and the effects of curing conditions, water/binder ratio, and ambient temperature on the rehydration are expressed by *D*_e0_ and *k_r_*. The degree of influence of each factor was characterized, and the model predicted the time required for the complete hydration of cement-based materials.

Based on the research results of Swaddiwudhipong et al. [104], the hydration products of Portland cement are chemically and physically close to the superposition of the hydration products of individual compounds under similar reaction conditions. Although hydration products may interact with each other and with other mineral compounds in a cement system, errors induced by the independent hydration assumption are low. Luo [40] regarded the complete hydration process of cement-based materials as the superposition of independent reactions of various cementitious components. As previously indicated, the rehydration process predominantly occurred in the latter stages of hydration. A new rehydration model was proposed in the present study, in accordance with the equation for the diffusion stage of cement hydration, as shown in Formula (7):(7)$$\alpha =1+\alpha_{0}-\sum_{i=1}^{4}(1-(K_{D}{}_{i}t{)}^{1/2}{)}^{3}$$
where *α* is the degree of hydration of cement; *α*_0_ is the initial hydration degree of cement at the beginning of rehydration; and *K*_Di_ denotes the rehydration reaction coefficient corresponding to the *i*-th cement component and characterizes the rehydration reaction rate of the i-th cement component.

The proposed rehydration model primarily considered four components, C_3_S, C_2_S, C_3_A, and C_4_AF, the corresponding *i* values of which were 1, 2, 3, and 4, respectively. The present authors contend that based on the variable values of *K*_Di_ under different water/binder ratios, rehydration temperatures, and curing environments, this hydration model could directly characterize the degree of influence of any single mineral component of a cement-based material on the hydration rate and predict with greater accuracy the degree of hydration of the pure cement systems under different water/binder ratio conditions.

Based on the hydration model [105], Liu [5] introduced the influence of rehydration to determine the degree of hydration during the hydration acceleration of cement-based materials under high-temperature water immersion conditions. He took the cement particles as a sphere and formed a square border during the hydration process, and considered the fineness of the cement particles. The effects on the cement hydration rate and the reduction in the contact area between cement particles and water within the square hydration units were considered to reflect the slowing down of the hydration process. A rehydration model was obtained from the above work. Elsewhere, Song [50] considered the diluting effect of fly ash to subdivide the hydration of cement-based materials in the cement-fly ash system into (i) a cement hydration component, and (ii) a fly ash hydration component, and considered the size effect of the sizes of cement and fly ash particles. A rehydration model was obtained as shown in Formula (8):(8)$$\left\{\begin{array}{l} \alpha =\gamma [(1-P)\alpha_{c}+P\cdot \alpha_{f}]\\ \alpha_{c}=\sum y_{c}(2R)\alpha_{cff}^{R}\\ \alpha_{f}=\sum y_{f}(2R)\alpha_{f}^{R} \end{array}\right.$$
where *α_c_* is the hydration degree of cement; *α*_f_ is the hydration degree of fly ash; *γ* is the interaction coefficient between fly ash and cement; $\alpha_{f}^{R}$ is the hydration degree of fly ash particles with a radius *R*_0_; $\alpha_{cff}^{R}$ is the hydration degree of cement after considering the dilution effect; and *y_f_*(2*R*) is the particle size distribution function of the cement particles.

As the hydration reaction proved to be extremely weak when the cement particle diameters exceeded 80 μm, the calculation range of the cement particle diameters was selected as 0–80 μm when analyzing the effect of the cement particle size distribution on the hydration process of cement in the present study [106]. The present authors considered their proposed model to be suitable for predicting the hydration degree in the rehydration process of the pure cement system and cement–fly ash system, and found that the simulation results obtained were consistent with the experimental results when the proportion of fly ash was below 30%, with a greater error value when the ratio of fly ash admixture exceeded 45%.

Based on Krstulović–Dabić’s cement hydration kinetics and cement hydration microscopic model [101,105], Liu [61] modified the cement-based material hydration model by considering the effects of water migration on the water/cement ratio and the influence of silica fume dosage, and introduced the coefficients of water/cement ratio influence, silica fume dosage influence, and rehydration effect to establish a multifactorial model of cement-based material rehydration, as shown in Formula (9):(9)$$\alpha_{i}=\gamma_{\mathrm{fh}}\xi_{\mathrm{wc}r}\xi_{sf}\sum_{2R=0}^{80}Y(2R)\alpha_{R,i}$$
where *ξ*_wcr_ is the influence coefficient of the water/binder ratio; *ξ*_sf_ is the influence coefficient of silica fume admixture; *γ*_fh_ is the influence coefficient of subsequent hydration; *Y*(2*R*) is the distribution function of cement particle size; and *α*_R,*i*_ is the degree of the hydration of cement particles at step *i* of the hydration process. The model was further modified to take into consideration the influence coefficient of rehydration. The influence coefficient relating to the silica fume admixture was summarized and offered a good applicability for pure cement systems and good predictive accuracy for cement–silica fume binary systems.

Liu [61] established a cement paste expansion strain prediction model after considering factors such as hydration temperature, moisture migration, and water/binder ratio, as shown in Formula (10):(10)$$\varepsilon_{FH}\left(t\right)=\frac{\gamma_{M}\gamma_{WCR}t}{a(T)+b(T)t}$$
where $\varepsilon_{FH}\left(t\right)$ is the expansion strain of the specimen; *t* is the hydration age; a(T) and b(T) are the functions relating to the water temperature *T*; $\gamma_{M}$ is the influence coefficient of water intrusion; and $\gamma_{WCR}$ is the influence coefficient of the water/binder ratio.

As the key difference between rehydration and initial hydration lies in the transfer of external water, it is essential to combine moisture transport data with the cement hydration model in the subsequent process of establishing a rehydration model. Although the above expansion strain model could only predict cement paste systems, its establishment process has the potential for it to be applied to other cement-based systems in the future, from which further similar expansion strain models can be established. Since the expansion strain prediction model expressed the relationship between the strain and hydration age, and the rehydration model expressed the relationship between the hydration degree and hydration age, the function of the hydration age represented by the degree of hydration could be obtained, from which the relationship between the expansion strain and hydration degree could in turn be established. The hydration degree corresponding to the maximum expansion strain during the rehydration process of cement-based materials under different water/binder ratios could be obtained from the curve of the expansion strain versus hydration degree. This could enable the damage risk of cement-based materials under rehydration to be assessed and controlled.

## 6. Conclusions

Cement-based materials with low water/binder ratios contain large amounts of unhydrated cement particles, and the rehydration of these unhydrated cement particles affects their macroscopic and microscopic properties. This study reviewed the existing research on rehydration and summarized the effects of rehydration on the cracks, chemically combined water, mechanical properties, and pore structure of cement-based materials. The specific conclusions are as follows:Rehydration is a process occurring in cement-based materials over prolonged periods: chemically combined water was found to further increase after 360 days of rehydration. It is therefore necessary to study the long-term performance of cement-based materials under rehydration. In its early stages, when the internal pores of the cement-based material are sufficient to accommodate rehydration products, rehydration can have a reinforcing effect. In the latter stages, however, when there are too few internal pores to accommodate new hydration products, internal stress is generated, which eventually further cracks;Up to a hydration age of 270 days, the compressive strength of cement-based materials increased continuously, but subsequently showed a decreasing trend, which is related to the effect of rehydration on the internal pore structure of cement-based materials. The capillary water absorption coefficient of cement-based materials up to a hydration age of 120 days continued to decrease, indicating that with the progress of rehydration, the content of capillary pores continued to decrease, and the internal density of the cement-based material continued to increase;The main products of rehydration were fibrous C-S-H gel and crystalline CaCO_3_. In addition, some amounts of Ca(OH)_2_ and AFt may be produced simultaneously. The influence of hydration products on porosity was closely related to the changes in the mechanical properties. Up to 270 days of hydration, the internal porosity of the cement-based materials was shown to continue to decrease, while the porosity appeared to increase after 270 days of hydration;As the water/binder ratio increased, the overall rehydration rate initially increased and then decreased. For cement-based materials with a water/binder ratio below 0.3, the increase in the rehydration temperature under the same water/binder ratio had an obvious promoting effect on the growth rate of chemically combined water. This effect of temperature on the rehydration rate was not obvious in the case of cement-based materials with a water/binder ratio of 0.3. In addition, the effects of mineral admixtures on rehydration varied according to different water/binder ratios;This study furthermore summarized the conditions of using different rehydration models described in the literature and proposed a method to establish the relationship between the expansion strain and the hydration degree under the rehydration of cement-based materials with low water/binder ratios. Furthermore, the hydration degree corresponding to the maximum expansion strain during the rehydration process of cement-based materials with low water/binder ratios was characterized. Finally, a method to evaluate and control the risk of damage to cement-based materials under rehydration was proposed.

## Figures and Tables

**Figure 1 materials-16-00970-f001:**
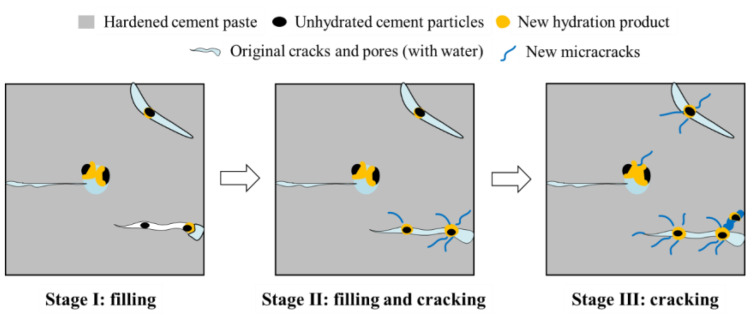
Rehydration repair mechanism.

**Figure 2 materials-16-00970-f002:**
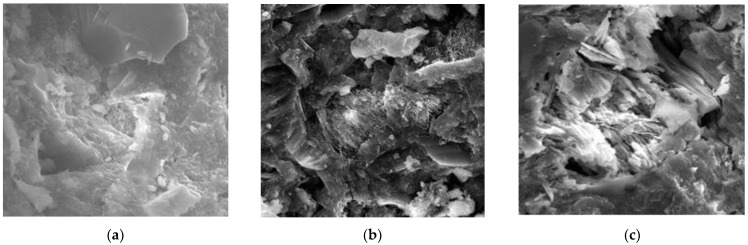
SEM morphology of cement-based materials under different rehydration periods [40]. (**a**) Rehydration for 28 days; (**b**) rehydration for 56 days; (**c**) rehydration for 180 days.

**Figure 3 materials-16-00970-f003:**
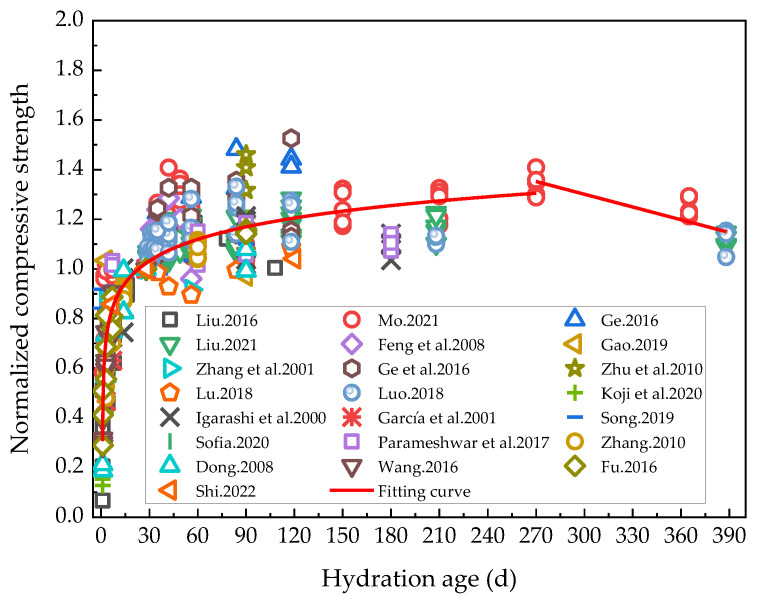
Variation curve of the normalized compressive strength with rehydration age [5,6,19,40,43,44,45,47,48,49,50,51,52,53,54,55,56,57,58,59,60,61].

**Figure 4 materials-16-00970-f004:**
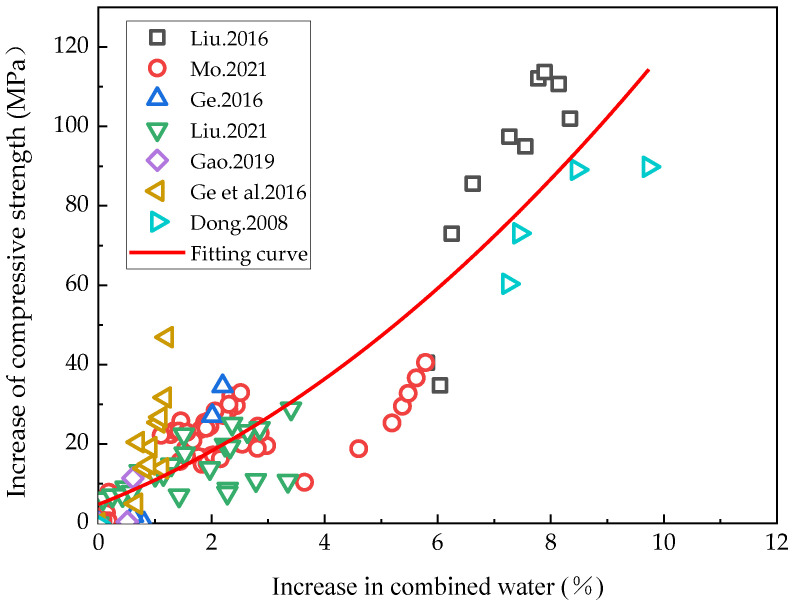
Relationship between the incremental combined water and compressive strength growth [5,45,55,56,57,60,61].

**Figure 5 materials-16-00970-f005:**
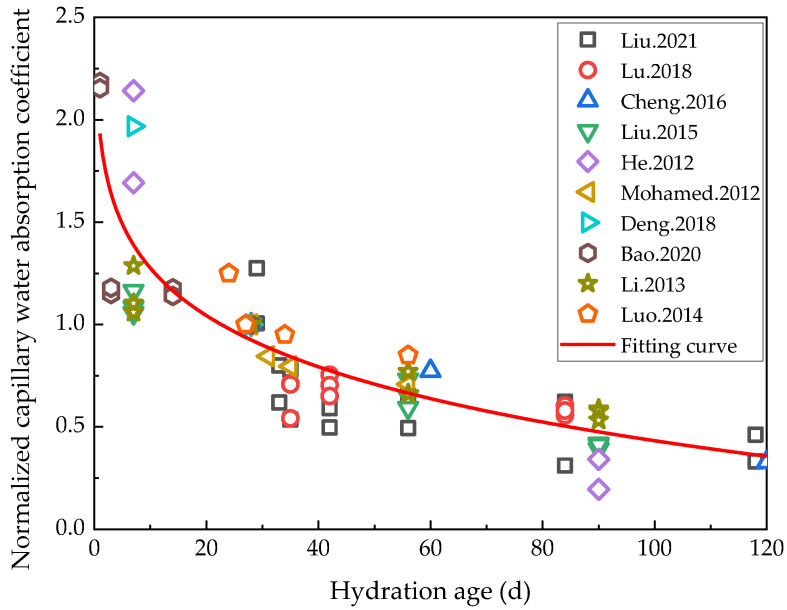
Relationship between the capillary absorption coefficient and hydration age [19,61,65,66,67,68,69,70,71,72].

**Figure 6 materials-16-00970-f006:**
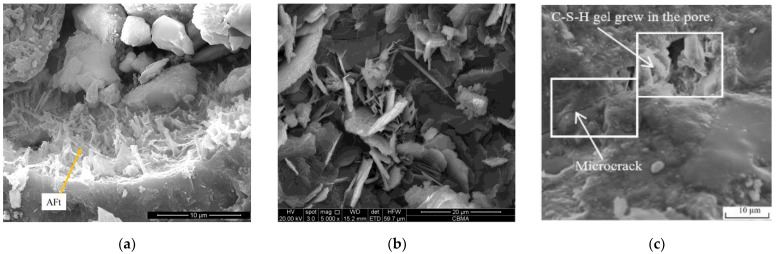
SEM images of the rehydration product morphology. (**a**) Ettringite. (**b**) Mixture of Ca(OH)_2_ and CaCO_3;_ (**c**) C-S-H gel.

**Figure 7 materials-16-00970-f007:**
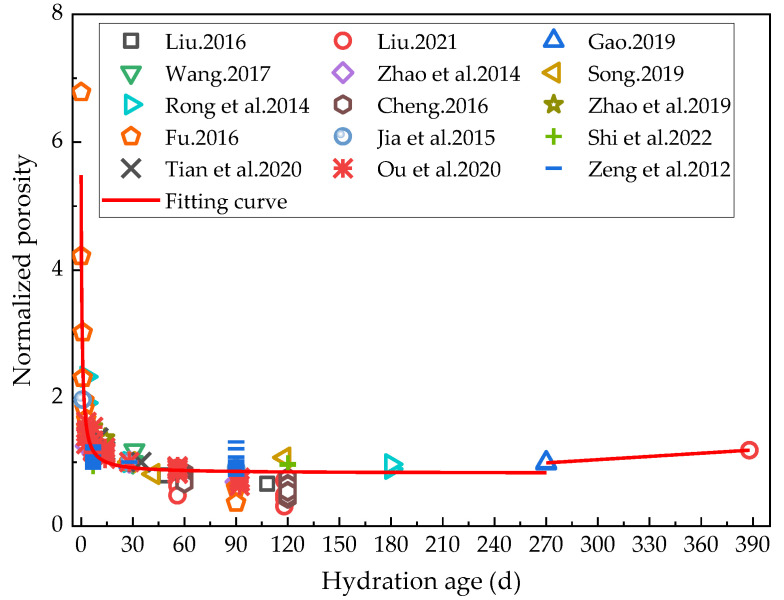
Relationship between porosity and hydration age [5,50,56,58,59,61,72,75,87,88,89,90,91,92,93].

**Figure 8 materials-16-00970-f008:**
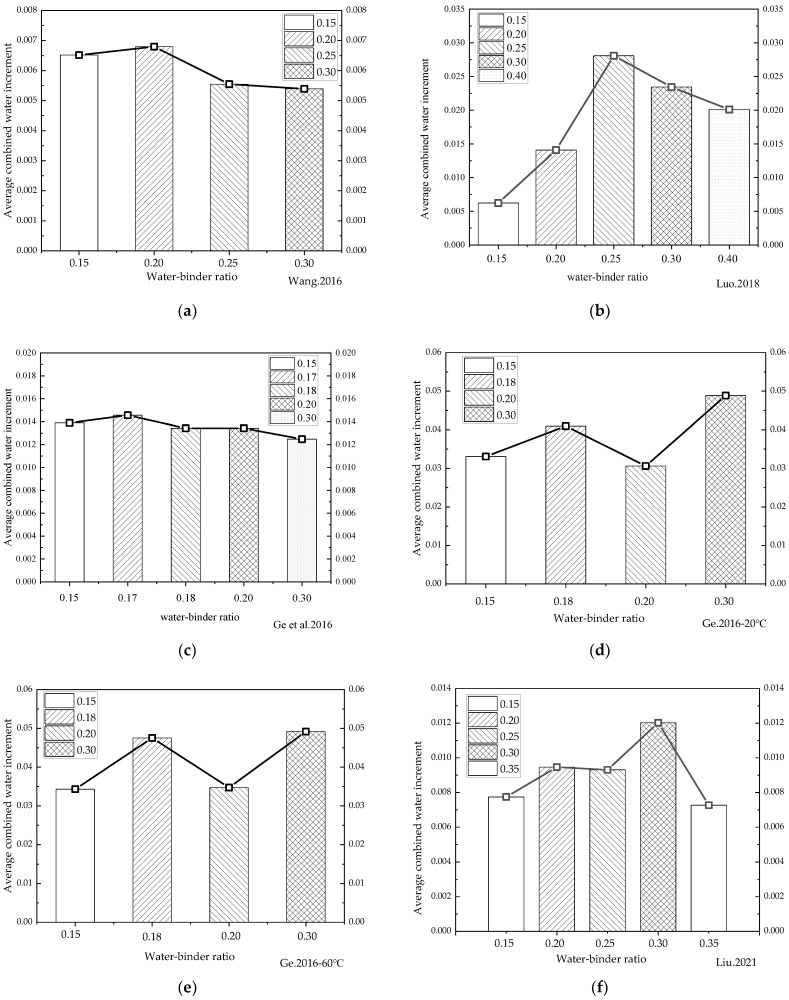
Variations in the average daily increment in combined water under different water/binder ratios. (**a**) [6] 180d average daily increment in combined water; (**b**) [40] 360d average daily increment in combined water; (**c**) [45] 90d average daily increment in combined water; (**d**) [60] 20 °C-90d average daily increment in combined water; (**e**) [60] 60 °C-90d average daily increment in combined water; (**f**) [61] 360d average daily increment in combined water.

**Figure 9 materials-16-00970-f009:**
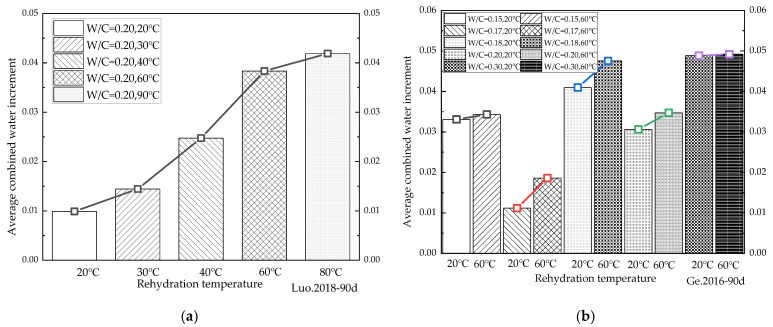
Variation in the average increment in the chemical combined water at different rehydration temperatures. (**a**) [40] 90d average growth rate at different rehydration temperatures; (**b**) [45] 90d average growth rate at different rehydration temperatures; (**c**) [50] 90d average growth rate at different rehydration temperatures; (**d**) [55] 28d average growth rate at different rehydration temperatures.

**Figure 10 materials-16-00970-f010:**
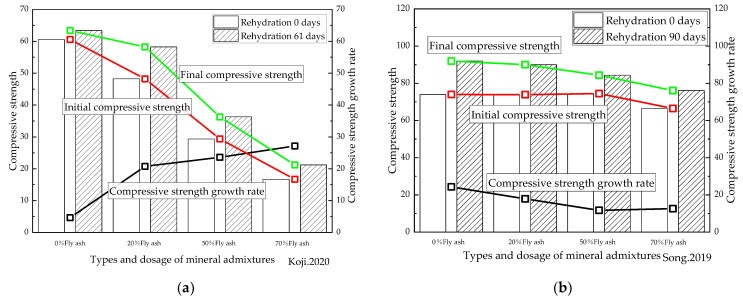
Relationship between the types and dosage of mineral admixtures and rehydration. (**a**) [48] (W/B = 0.4); (**b**) [50] (W/B = 0.3); (**c**) [53] (W/B = 0.35); (**d**) [55] (W/B = 0.2); (**e**) [60] (W/B = 0.17); (**f**) [61] (W/B = 0.2). Me is metakaolin, Li is limestone powder, and FA is fly ash.

**Figure 11 materials-16-00970-f011:**
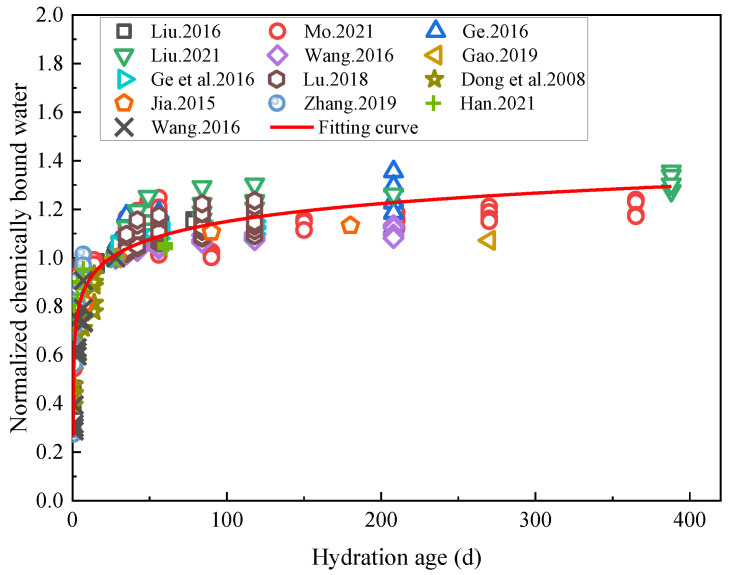
Variation curve of the chemically bound water content with hydration age after normalization [5,6,19,45,55,56,57,60,61,92,98,99,100].

**Table 1 materials-16-00970-t001:** Repairable crack widths by rehydration of the cement-based materials.

	W/B Ratio	Age/d	Repairable Crack Width/μm	Refs.
Aldea et al.	0.45	100	200–300	[29]
Tomczak and Jakubowski	0.28	28	388	[30]
	0.23		460	
Ahn and Kishi	0.45	3	150	[32]
		7	160–220	
Yang et al.	0.3	-	150	[35]
Benny et al.	0.28	21	40–75	[36]
Liu et al.	0.25	120	150	[37]

**Table 2 materials-16-00970-t002:** Influence of rehydration on the macroscopic properties of cement-based materials.

Macroscopic Properties	Change Pattern under Rehydration
Cracks	Whether or not cracks are generated by rehydration within structures depends on whether or not there is space for rehydration products within these.
Mechanical properties	First increase then decrease.
Combined water	Increasing hydration of cement-based materials
Permeability	The permeability of cement-based materials decreases as the age of rehydration increases in the early stage of rehydration

**Table 3 materials-16-00970-t003:** Rehydration products.

Rehydration Products	Refs.
CaCO_3_	[21,34,77]
Ca(OH)_2_, C-S-H gel and CaCO_3_	[22]
C-S-H gel, Ca(OH)_2_, ettringite, and CaCO_3_	[37]
Ca(OH)_2_, C-S-H gel	[74]
C-S-H gel	[75,81]
C-S-H gels, Ca(OH)_2_ and AFt	[76]
Ca(OH)_2_, CaCO_3_	[80]
C-S-H gel, CaCO_3_	[64]

**Table 4 materials-16-00970-t004:** Influence of different factors on rehydration.

Factor	Rehydration Rate Change Pattern
W/B ratio	The overall trend is to increase and then decrease
Rehydration temperature	As the rehydration temperature increases, the rehydration rate keeps increasing
Mineral admixture	(a) When the dosage of metakaolin is low, it will promote the repair and enhancing effect of cement-based materials under rehydration; when the dosage of metakaolin is high, it will hinder the repair and enhancing effect of cement-based materials under rehydration; (b) The addition of limestone powder will facilitate the repair and enhancing effect of cement-based materials under rehydration; (c) For cement-based materials with low water/binder ratios, the addition of silica fume will not be conducive to the repair and enhancing effect under rehydration; for cement-based materials with slightly larger water/binder ratios, the addition of silica fume will promote the repair and enhancing effect under rehydration; (d) For cement-based materials with low water/binder ratios, the addition of fly ash is not conducive to the repair and enhancing effect under rehydration; for cement-based materials with high water/binder ratios, the addition of fly ash will improve the repair and enhancing effect under rehydration; (e) The addition of slag will improve the repair and enhancing effect under rehydration.

## Data Availability

The data supporting the findings of the study are available within the article.

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
