# Peer review of "The Influence of Rehydration on the Properties of Portland Cement-Based Materials with Low Water/Binder Ratios: A Review of Existing Research"

_materials, 2023, doi:10.3390/ma16030970_

Round 1
Reviewer 1 Report
The paper "The influence of rehydration on the properties of cement-based materials with low water/binder ratios: A review of existing research" presents a relevant theme and within the scope of this journal, and can be considered after some corrections suggested below:
(a) The abstract is generally well written, however in terms of content it is generic, i.e., the authors lack an in-depth study of the quantitative results of this research;
(b) Scientific innovation is limited in the introduction of the paper, the authors must go deeper and detail what this research differs from countless others that exist on this topic, this must be evidenced together with the objectives at the end of the introduction;
(c) The state of the art of the evaluated topic needs to be improved by the authors, note that some topics are absent and need to be known with current research, such as: 10.1016/j.cscm.2021.e00714; 10.1016/j.cscm.2021.e00727; 10.1016/j.conbuildmat.2017.10.111.
(d) In some passages, the authors must be more critical, and propose advances in the state of the art of the subject, this proves to be limited in many issues addressed, making a paper a simple compilation of results;
(e) Topic 6 "6. Conclusion and outlook” is very long, it is subdivided, and the authors should rethink the way of their presentation, note that the conclusion should always be at the end and short and objective to the readers, obviously future suggestions and gaps in the literature are important to be added in other topics.
Author Response
Response to the reviewer's comments
January 14, 2023
Dear Editor and Reviewers:
Thank you for your letter and comments concerning our manuscript entitled “The influence of rehydration on the properties of cement-based materials with low water/binder ratios: A review of existing research” (ID: materials-2108953). We are very grateful for your help. We have revised the manuscript carefully based on the comments and consulted professional editors to improve the quality of language. The revisions made to our manuscript and our responses to the reviewers’ comments are as follows. At the same time, we have uploaded the original manuscript with revision marks as attachments for you to review.
Comments: The abstract is generally well written, however in terms of content it is generic, i.e., the authors lack an in-depth study of the quantitative results of this research.
Response: The authors are very grateful to the reviewer for these comments. They are very important for our research. I have added several quantitative analyses to the article and summarized the corresponding statements of the corresponding chapters using tables in the appropriate places. The added content is as follows:
Line-251: “The normalized compressive strength of cement-based materials was reduced by approximately 15.21% under rehydration. Current studies on rehydration of cement-based materials for more than 150 days are scarce. In order to fully study the effect of long-term rehydration on the properties of cement-based materials, in the future, it is necessary to investigate the performance changes of cement-based materials after the rehydration age exceeds 150 days.”
Line-320: “The normalized capillary absorption coefficient of cementitious materials with the hydration age of 120d was reduced to 18.65% of the initial state.”
Line-531: “Figure 9(a) shows that after 90 days of rehydration, the average daily increments of chemically combined water at rehydration temperatures of 40 °C, 60 °C, and 90 °C increased by 27.20%, 34.49% and 53.93%, respectively, compared with that at rehydration temperature of 20 °C for cement-based materials with a water/binder ratio of 0.3. Figure 9(b) indicates that after 90 days of rehydration for cement-based materials with a water/binder ratio of 0.2, the average daily increments of chemically combined water at rehydration temperatures of 30 °C, 40 °C, 60 °C, and 90 °C increased by 31.51%, 60.06%, 74.20%, and 76.39%, respectively, compared with that at rehydration temperature of 20 °C. Figure 9(c) illustrates that the average daily increment of chemically combined water at a rehydration temperature of 80 °C was 20.88 times higher than that at a rehydration temperature of 20 °C for cement-based materials with a water/binder ratio of 0.2 after 28 days of rehydration. As shown in Figure 9(d), after 90 days of rehydration of cement-based materials with water/binder ratios of 0.15, 0.17, 0.18, 0.20, and 0.30, respectively, the average daily increment of chemically combined water at a rehydration temperature of 60 °C was 3.69%, 66.37%, 16.09%, 13.35%, and 0.70% higher than that at a rehydration temperature of 20 °C. The cement-based material with a water/binder ratio of 0.2 had the largest increase in the average daily increment of chemically combined water under rehydration. This indicates that for cement-based materials with a water/binder ratio of 0.2, more attention should be paid to the adverse effect of rehydration reactions on them.”
The tables added can be found in Tables 2, 3, and 4 in the article.
Comments: Scientific innovation is limited in the introduction of the paper, the authors must go deeper and detail what this research differs from countless others that exist on this topic, this must be evidenced together with the objectives at the end of the introduction.
Response: Thank you for this important suggestion. The differences between what is studied in this paper and other existing papers have been redetailed by us, and correspond to the order of the conclusions of the paper. The revised content is located in the last paragraph of the Introduction, which reads as follows: “Many cement-based materials with a low water/binder ratio are subject to significant rehydration reactions due to the presence of large amounts of unhydrated cementitious materials within them. Notwithstanding the above, few studies have been conducted to date focusing on the rehydration of cement-based materials with a low water/binder ratio, and on the effect of rehydration on their microscopic properties, as this effect is mainly reflected in their macroscopic properties. There is no consensus on the long-term performance evolution law in controlling the adverse effect of rehydration reaction on cement-based materials with a low water/binder ratio, and there lacks an effective damage risk evaluation and control method. Since the rehydration process of cement-based materials with a low water/binder ratio and its mechanism remain unclear, it is necessary to study the mechanism of performance change in cement-based materials with a low water/binder ratio under the rehydration effect. To this end, this study was conducted. Firstly, the effect of rehydration on the long-term performance of cement-based materials with a low water/binder ratio was systematically analyzed, and the repair of cracks, mechanical properties, and permeability changes in cement-based materials under rehydration were systematically analyzed. Secondly, the influencing factors of rehydration were determined, aiming to provide theoretical guidance for the long-term performance stability of cementitious materials, and to clarify how to effectively control the adverse effect of rehydration on cement-based materials with a low water/binder ratio. Finally, a systematic solution was provided to suppress the deterioration caused by rehydration on cement-based materials.”
Comments: The state of the art of the evaluated topic needs to be improved by the authors, note that some topics are absent and need to be known with current research, such as: 10.1016/j.cscm.2021.e00714; 10.1016/j.cscm.2021.e00727; 10.1016/j.conbuildmat.2017.10.111.
Response: Thank you for your comment. I have cited your recommended referenceliterature related to this paper, for details please see reference 45 and reference 94 in the corresponding citation section. We have studied some of the methods mentioned in the uncited articles and plan to adopt them in the subsequent studies Details of the additions are as follows:
Line222: “Roz et al. [45] studied the mechanical properties of wood fiber cement, and the results showed that the presence of wood fibers in cement-based materials doped with wood fibers would not be conducive to improving the mechanical properties of cement-based materials under rehydration. It is mainly due to the reversible damage caused by water to the hydrogen bond between the cellulose fibers and the cement-based matrix, which eventually leads to a significant decrease in the flexural properties of cementitious materials.”
Line568: “Hee et al. [94] studied the properties of Ferronickel slag (FNA)-doped cement-based materials by using XRD, TG analysis, and MIP. They found that the average pore size of FNA-doped cement-based materials was reduced, and that the rehydration products produced by the reaction between calcium hydroxide and internal moisture from the decomposition of C-S-H gels at high temperatures were different from those produced by the rehydration products of cement-based materials not doped with FNA and could have better refractory properties.”
Comments: In some passages, the authors must be more critical, and propose advances in the state of the art of the subject, this proves to be limited in many issues addressed, making a paper a simple compilation of results.
Response: Thank you for your comment. I have made some additions to the article for some paragraphs that are not expressed in detail. The experimental methods of the current study are summarized and some more novel ones that can be used for the corresponding experiments are added. The detailed additions are as follows:
Line327: “The changes in macroscopic properties of cement-based materials under rehydration are summarized in Table 2. The current capillary water absorption test results are often largely dispersed. In the future study, it is recommended to use low-field nuclear magnetic resonance (NMR) and X-ray computed tomography (X-CT) methods to detect the capillary water absorption and capillary water absorption depth, so that the accuracy of characterizing the permeability of cement-based materials can be improved.”
Line410: “The current commonly used method for detecting rehydration products is XRD. The XRD results obtained can be combined with transmission electron microscopy to analyze the type of rehydration products and with thermogravimetry to analyze their content, thus achieving a more accurate quantitative analysis.”
Line458: “Currently, the pores of cement-based materials are often tested and analyzed by MIP, which requires the destruction of the specimen. In order to ensure the continuity of pore structure changes during the rehydration of cement-based materials, the low-field NMR method can be used to test the pore structure of cement-based materials. The results obtained by low-field NMR can be compared with those obtained by the MIP method to characterize the pore structure of cement-based materials under rehydration more clearly.”
Line686: “Current methods for testing the hydration of cement-based materials often use the method of testing the chemically combine water content, and the test samples may have large errors due to different sampling locations. The relationship between resistivity and hydration can be used in subsequent studies to achieve a nondestructive method for testing the hydration of cement-based materials.”
We re-examined the data used in the article and re-analyzed the results.
Comments: Topic 6"6. Conclusion and outlook” is very long, it is subdivided, and the authors should rethink the way of their presentation, note that the conclusion should always be at the end and short and objective to the readers, obviously future suggestions and gaps in the literature are important to be added in other topics.
Response: The conclusion section of the article has been resummarized and placed in a separate section at the end of the article, and the revised content is shown below: “Cement-based materials with low water/binder ratios contain large amounts of unhydrated cement particles, and the rehydration of these unhydrated cement particles affects their macroscopic and microscopic properties. This study reviewed the existing research on rehydration and summarized the effects of rehydration on cracks, chemically combined water, mechanical properties, and pore structure of cement-based materials. The specific conclusions are as follows:
- Rehydration is a process occurring in cement-based materials over prolonged periods: chemically combined water has been found to further increase after 360 days’ rehydration. It is therefore necessary to study the long-term performance of cement-based materials under rehydration. In its early stages, when the internal pores of the cement-based material are sufficient to accommodate rehydration products, rehydration can have a reinforcing effect. In the latter stages, however, when there are too few internal pores to accommodate new hydration products, internal stress is generated, which eventually further cracks;
- Up to a hydration age of 270 days, the compressive strength of cement-based materials increases continuously, but subsequently shows a decreasing trend, which is related to the effect of rehydration on the internal pore structure of cement-based materials. The capillary water absorption coefficient of cement-based materials up to a hydration age of 120 days continues to decrease, indicating that with the progress of rehydration, the content of capillary pores continues to decrease, and the internal density of the cement-based material continues to increase;
- The main products of rehydration are fibrous C-S-H gel and crystalline CaCO3. In addition, some amounts of Ca(OH)2 and AFt may be produced simultaneously. The influence of hydration products on porosity is closely related to the changes in mechanical properties. Up to 270 days’ hydration, the internal porosity of the cement-based materials is shown to continue to decrease, while the porosity appears to increase after 270 days’ hydration;
- As the water/binder ratio increases, the overall rehydration rate initially increases and then decreases. For cement-based materials with a water/binder ratio below 0.3, the increasing of rehydration temperature under the same water/binder ratio, has an obvious promoting effect on the growth rate of chemically combined water. This effect of temperature on rehydration rate was not obvious in the case of cement-based materials with a water/binder ratio of 0.3. In addition, the effects of mineral admixtures on rehydration varies according to different water/binder ratios;
- This study furthermore summarizs the conditions of using different rehydration models described in the literature and proposes a method to establish the relationship between the expansion strain and the hydration degree under the rehydration of cement-based materials with low water/binder ratios. Furthermore, the hydration degree corresponding to the maximum expansion strain during the rehydration process of cement-based materials with low water/binder ratios is characterized. Lastly, a method to evaluate and control the risk of damage to cement-based materials under rehydration is proposed.”
Reviewer 2 Report
This study provides some insights regarding the review on the influence of rehydration on the properties of cement-based materials with low water/binder ratios. The arrangement of subsection and important summary have been highlighted. However, the depth of discussion and critics is still needed. Most of the references are up to date. My comments as listed below:
1. It is much significant if each section can be summarised in the Table, based on the finding from previous research, then followed by the summary for each section.
2. Figure 9: All data comes from the same types of cement? If not, it should be discussed, which type gave the most influent in rehydration.
3. Section 4.2: The critics should be enhanced and cleared.
4. Is it possible to measure the percentage of rehydration in cement/concrete?
Author Response
Response to the reviewer's comments
January 14, 2023
Dear Editor and Reviewers:
Thank you for your letter and comments concerning our manuscript entitled “The influence of rehydration on the properties of cement-based materials with low water/binder ratios: A review of existing research” (ID: materials-2108953). We are very grateful for your help. We have revised the manuscript carefully based on the comments and consulted professional editors to improve the quality of language. The revisions made to our manuscript and our responses to the reviewers’ comments are as follows. At the same time, we have uploaded the original manuscript with revision marks as attachments for you to review.
Comments: 1-It is much significant if each section can be summarised in the Table, based on the finding from previous research, then followed by the summary for each section.
Response: Thank you for this important comment. I have summarized the expressions in the necessary places and added Tables 2, 3 and 4 to the article, the details of the table are as follows, thank you again for your suggestions.
Line333:Table 2. Influence of rehydration on macroscopic properties of cement-based materials
Macroscopic properties |
Change pattern under rehydration |
Cracks |
Whether or not cracks are generated by rehydration within structures depends on whether or not there is space for rehydration products within these. |
Mechanical properties |
First increase then decrease. |
Combined water |
Increasing hydration of cement-based materials |
Permeability |
The permeability of cement-based materials decreases as the age of rehydration increases in the early stage of rehydration |
Line407:Table 3. Rehydration products
Rehydration products |
Ref. |
CaCO3 |
[21,34,71] |
Ca(OH)2, C-S-H gel and CaCO3 |
[22] |
C-S-H gel, Ca(OH)2, ettringite, and CaCO3 |
[37] |
Ca(OH)2, C-S-H gel |
[68] |
C-S-H gel |
[69,76] |
C-S-H gels, Ca(OH)2 and AFt |
[70] |
Ca(OH)2, CaCO3 |
[74] |
C-S-H gel, CaCO3 |
[75] |
Line407:Table 4. Influence of different factors on rehydration
Factor |
Rehydration rate change pattern |
W/B ratio |
The overall trend is to increase and then decrease |
Rehydration temperature |
As the rehydration temperature increases, the rehydration rate keeps increasing |
Mineral admixture |
a) When the dosage of metakaolin is low, it will promote the repair and enhancing effect of cement-based materials under rehydration; when the dosage of metakaolin is high, it will hinder the repair and enhancing effect of cement-based materials under rehydration b) The addition of limestone powder will facilitate the repair and enhancing effect of cement-based materials under rehydration c) For cement-based materials with low water/binder ratios, the addition of silica fume will not be conducive to the repair and enhancing effect under rehydration; for cement-based materials with slightly larger water/binder ratios, the addition of silica fume will promote the repair and enhancing effect under rehydration d) For cement-based materials with low water/binder ratios, the addition of fly ash is not conducive to the repair and enhancing effect under rehydration; for cement-based materials with high water/binder ratios, the addition of fly ash will improve the repair and enhancing effect under rehydration e) The addition of slag will improve the repair and enhancing effect under rehydration |
Comments: 2-Figure 9: All data comes from the same types of cement? If not, it should be discussed, which type gave the most influent in rehydration.
Response: Thank you for your comment. The cement types discussed throughout the paper are all Portland cements, and no other cement types exist.
Comments: 3-Section 4.2: The critics should be enhanced and cleared.
Response: Thank you for this important suggestion. In this section I have provided additional explanations and quantitative analysis on the average daily chemically bound water increments. The additions are as follows:
Line-511: “Figure 9(a) shows that after 90 days of rehydration, the average daily increments of chemically combined water at rehydration temperatures of 40 °C, 60 °C, and 90 °C increased by 27.20%, 34.49% and 53.93%, respectively, compared with that at rehydration temperature of 20 °C for cement-based materials with a water/binder ratio of 0.3. Figure 9(b) indicates that after 90 days of rehydration for cement-based materials with a water/binder ratio of 0.2, the average daily increments of chemically combined water at rehydration temperatures of 30 °C, 40 °C, 60 °C, and 90 °C increased by 31.51%, 60.06%, 74.20%, and 76.39%, respectively, compared with that at rehydration temperature of 20 °C. Figure 9(c) illustrates that the average daily increment of chemically combined water at a rehydration temperature of 80 °C was 20.88 times higher than that at a rehydration temperature of 20 °C for cement-based materials with a water/binder ratio of 0.2 after 28 days of rehydration. As shown in Figure 9(d), after 90 days of rehydration of cement-based materials with water/binder ratios of 0.15, 0.17, 0.18, 0.20, and 0.30, respectively, the average daily increment of chemically combined water at a rehydration temperature of 60 °C was 3.69%, 66.37%, 16.09%, 13.35%, and 0.70% higher than that at a rehydration temperature of 20 °C. The cement-based material with a water/binder ratio of 0.2 had the largest increase in the average daily increment of chemically combined water under rehydration. This indicates that for cement-based materials with a water/binder ratio of 0.2, more attention should be paid to the adverse effect of rehydration reactions on them.”
Comments: 4-Is it possible to measure the percentage of rehydration in cement/concrete?
Response: Thanks for this important comment. Currently, the percentage growth of hydration of cement-based materials under rehydration is generally determined by the change in chemically bound water content. Our team has been trying to study the growth rate of hydration of cement-based materials under rehydration by resistivity and low-field NMR methods, which will be shown in the subsequent research results of our team.
Reviewer 3 Report
The article is devoted to the effect of rehydration on the properties of cement composites. The article is well structured and organized. In my opinion, there are some points that need improvement:
1) in the title of the article it is necessary to indicate not on the basis of "cement", in "Portland cement", since other types of cements: Romancement, Sorel cement, aluminous cement are not considered in the article;
2) references to the literature used are presented in ascending order. However, link #33 is followed by link #72 (line 128), and so on.
3) lines No. 208-210 delete;
4) what water/binder ratio do the authors consider to be low? The article presents studies on cement pastes with a ratio of water / binder = 0.4, which is not low. It needs to be clarified.
Author Response
Response to the reviewer's comments
January 14, 2023
Dear Editor and Reviewers:
Thank you for your letter and comments concerning our manuscript entitled “The influence of rehydration on the properties of cement-based materials with low water/binder ratios: A review of existing research” (ID: materials-2108953). We are very grateful for your help. We have revised the manuscript carefully based on the comments and consulted professional editors to improve the quality of language. The revisions made to our manuscript and our responses to the reviewers’ comments are as follows. At the same time, we have uploaded the original manuscript with revision marks as attachments for you to review.
Comments: 1-In the title of the article it is necessary to indicate not on the basis of "cement", in "Portland cement", since other types of cements: Romancement, Sorel cement, aluminous cement are not considered in the article;
Response: Thank you for this important suggestion, I have changed "cement" to " Portland cement."
Comments: 2-References to the literature used are presented in ascending order. However, link #33 is followed by link #72 (line 128), and so on.
Response: Thank you for your comment. I have carefully reordered the references cited in the article."
Comments: 3-Lines No. 208-210 delete;
Response: I am sorry for the mistake. And I have deleted lines 208 to 210.
Comments: 4-What water/binder ratio do the authors consider to be low? The article presents studies on cement pastes with a ratio of water / binder = 0.4, which is not low. It needs to be clarified.
Response: According to powers' theory, when the water/binder ratio of cement-based materials is lower than 0.38, there will be a large number of unhydrated cement particles inside the cement-based materials during the service phase, when it encounters water again, a rehydration reaction will occur. The low water/binder ratio cement-based material in this paper is broadly defined as a cement-based material capable of having unhydrated cement particles present internally during the service phase. And for the data in the literature cited in the text with a water/binder ratio of 0.4, although the water/binder ratio in its text is 0.4, there are still unhydrated cement particles inside the cement-based material after curing is completed, so the article is cited here to show that rehydration does not only exist inside cementitious materials with low water/binder ratios in the traditional sense, but also occurs inside cementitious materials for larger water-cement ratios. I have added the relevant additional notes to the article.